# Voltammetric Determination of the Total Content of the Most Commonly Occurring Estrogens in Water Media

**DOI:** 10.3390/molecules30030751

**Published:** 2025-02-06

**Authors:** Jaromíra Chýlková, Jan Bartáček, Natálie Měchová, Miloš Sedlák, Jiří Váňa

**Affiliations:** 1Institute of Environmental and Chemical Engineering, Faculty of Chemical Technology, University of Pardubice, Studentská 573, CZ 532 10 Pardubice, Czech Republic; jaromira.chylkova@upce.cz (J.C.);; 2Institute of Organic Chemistry and Technology, Faculty of Chemical Technology, University of Pardubice, Studentská 573, CZ 532 10 Pardubice, Czech Republic; milos.sedlak@upce.cz (M.S.); jiri.vana@upce.cz (J.V.)

**Keywords:** estrone, estradiol, estriol, ethinylestradiol, differential pulse voltammetry, boron-doped diamond electrode

## Abstract

Estrogens in aquatic environments pose significant ecological and health risks due to their cumulative effects rather than individual impacts. This study investigates the voltammetric behavior of estrone (E1), 17β-estradiol (E2), estriol (E3), and 17α-ethinylestradiol (EE2), presenting a cost-effective and straightforward method for their simultaneous determination. Using differential pulse voltammetry (DPV) with a boron-doped diamond electrode, the method demonstrates high precision (deviations under 4%) and a linear dynamic range of 15.35–134.55 µmol·L^−1^. Integration of a vacuum evaporation step reduced detection limits to 10^−8^ mol·L^−1^, enabling effective analysis of real water samples. This optimized approach ensures practical applicability for monitoring total estrogen content in aquatic systems, providing an accessible and reliable alternative to conventional methods.

## 1. Introduction

In recent years, estrogens have garnered significant attention due to their increasing concentrations in soil and aquatic environments. This concern arises from their potential environmental and human health impacts [1]. Estrogens, a class of steroid hormones, are critical regulators of reproductive and endocrine systems in animals. These compounds enter water bodies from various sources, such as wastewater treatment plants, agricultural runoff, industrial discharges, municipal sewage, and pharmaceutical products. The predominant source of estrogenic substances in urban wastewater is human excretion, primarily in the form of estrone (E1), 17β-estradiol (E2), estriol (E3), and, in the case of hormonal contraceptive use, 17α-ethinylestradiol (EE2) (Figure 1) [2].

Ethinylestradiol, present in wastewater at concentrations ranging from 10^1^ ng·L^−1^ to 10^2^ ng·L^−1^, is considered the major contributor to the overall estrogenic activity of these waters [3]. Livestock farming also significantly contributes to the levels of these steroidal estrogens in aquatic systems [4,5].

Estrogens in aquatic environments pose a threat to aquatic organisms by disrupting their endocrine systems, leading to reproductive and developmental abnormalities. Long-term human exposure to low estrogen concentrations is associated with disturbances in hormonal balance and an increased risk of breast cancer and other serious health issues [6,7,8,9,10,11].

In analytical practice, the most widely used technique for determining estrogens in water is liquid chromatography–mass spectrometry (LC-MS), as reported in several studies [12,13,14]. While highly sensitive, LC-MS is expensive, requires specialized instrumentation, and involves labor-intensive sample preparation.

The development of new voltammetric methods for estrogen determination in water offers a promising alternative, providing more cost-effective and accessible monitoring techniques, while also aiding in the identification of pollution sources. Several electrochemical approaches for estrogen detection have been explored in the literature, often utilizing modified electrodes and various electrochemical environments to achieve high sensitivity [15,16,17,18,19,20,21,22,23,24,25,26,27,28]. However, most studies target the detection of individual estrogenic compounds. Since these compounds exert a cumulative effect in aquatic environments, determining their total concentration is crucial.

This study focuses on the voltammetric determination of four prevalent estrogens—estrone, 17β-estradiol, estriol, and ethinylestradiol—and presents optimized conditions for their simultaneous quantification. The boron-doped diamond electrode (BDDE), utilized in this work, offers significant advantages for modern electroanalytical chemistry. The BDDE is characterized by a wide potential window in water-organic solvents, ranging approximately from −1000 mV to 2500 mV, which is crucial for the detection of compounds like estrogens that are generally poorly oxidizable and require high potentials for their electrochemical detection. Additionally, its low background current, high mechanical resistance, and excellent chemical stability make it an ideal tool for sensitive and reliable detection of these analytes [29,30,31].

## 2. Results and Discussion

### 2.1. Electrochemical Behavior of 17α-Ethinylestradiol, Estrone, 17β-Estradiol, and Estriol

Given the structures of the analyzed estrogenic compounds (Figure 1), their electrochemical oxidation was investigated using differential pulse voltammetry (DPV) as the selected anodic method. Initially, 17α-ethinylestradiol (EE2) was chosen as a representative analyte due to its relevance and abundance in environmental matrices. EE2 was selected not only for its widespread occurrence and relevance in environmental samples but also due to its structural characteristics, which make it a suitable model compound for optimizing the electrochemical conditions for all four estrogens.

Experiments examining the composition of the supporting electrolyte, including its pH and the proportion of organic solvent, revealed that purely aqueous environments provide anodic oxidation peaks at lower positive potentials. However, as the EE2 concentration increased, these peaks shifted from approximately +975 mV to +1030 mV, and linearity was achieved only within a narrower concentration range(Table 1). Introducing acetonitrile (ACN) into the electrolyte shifted the peak potentials toward more positive values, with the magnitude of the shift increasing alongside the proportion of ACN. For instance, at 33% ACN, peak potentials ranged from +1220 mV to +1270 mV, while at 66% ACN they were between +1430 mV and +1475 mV.

Linear regression analysis (Table 1) demonstrated that, within the concentration range of 15.35–134.55 µmol·L^−1^, the presence of ACN significantly enhanced linearity, as evidenced by correlation coefficients close to unity. Although increasing ACN content reduced the sensitivity (slope) slightly, the overall linearity improved. Based on these findings, a supporting electrolyte containing Britton–Robinson (BR) buffer at pH 9.04 with 33% ACN was selected for further analyses. This compromise offered a suitable balance between linearity and sensitivity.

### 2.2. Effect of pH on EE2 Oxidation

The pH of the BR buffer was varied from 2.04 to 10.00 to identify optimal conditions for EE2 determination (see Appendix A). Under acidic conditions (pH 2.04), the oxidation peaks shifted significantly toward more positive potentials with increasing EE2 concentration (from +1730 mV at 15.35 µmol·L^−1^ to +1910 mV at 134.55 µmol·L^−1^), and pronounced nonlinearity was observed (Table 2). At pH values of 4.03 and 6.06, the oxidation occurred within +1140 to +1330 mV, but these conditions again yielded nonlinear current–concentration relationships.

In alkaline media (pH 9.04 and 10.00), the oxidation process became more complex, with two distinct peaks emerging (Appendix A). For pH 9.04, the first peak appeared between +1215 and +1270 mV and the second between +1625 and +1670 mV. At pH 10.00, the corresponding peaks occurred at +1240 to +1290 mV and +1610 to +1675 mV, respectively. Despite this complexity, selecting the first oxidation peak for quantitative evaluation provided a satisfactory linear response and was easier to evaluate compared to the second peak. The calibration equations at pH 9.04 and 10.00 were *I* = 1.86*c* + 8.39 and *I* = 1.78*c* + 8.87, respectively (where *I* is in nA and *c* is in µmol·L^−1^), as shown in Table 2. The pH 9.04 condition ultimately offered a better sensitivity, making it the preferred choice for subsequent analyses.

### 2.3. Quantitative Determination of EE2 Under Optimized Conditions

Based on all these considerations, the optimized conditions were established as BR buffer at pH 9.04 with 33% ACN. Under these conditions, initial experiments demonstrated a linear dynamic range of 15.35–134.55 µmol·L^−1^ for EE2, providing excellent linear correlation and adequate sensitivity within the studied concentration range (Figure 2). Additional tests at lower concentrations (0.94–4.7 µmol·L^−1^) further confirmed the high sensitivity of the method, resulting in an LOD of 0.08 µmol·L^−1^ and an LOQ of 0.27 µmol·L^−1^ (see Appendix A). These results validate the applicability of the optimized conditions for detecting very low EE2 concentrations.

### 2.4. Simultaneous Analysis of Estrone, 17β-Estradiol, Estriol, and EE2

After establishing suitable conditions for EE2, the method’s applicability was extended to estrone (E1), 17β-estradiol (E2), and estriol (E3). All these compounds exhibited characteristic oxidation peaks under the selected conditions (Figure 3). For quantitative evaluation, using the first peak was preferable, as it provided higher and more reproducible currents. Estrone and E2 produced first peaks at identical potentials around +1200 mV, whereas EE2 and E3 peaks appeared at +1240 mV and +1280 mV, respectively.

The optimized conditions developed for EE2 were seamlessly adapted to E1, E2, and E3, enabling the preparation of concentration series and the generation of calibration curves (Appendix A). This approach enabled the identification of linear current responses for all three compounds, confirming the applicability of the optimized conditions for quantitative evaluation.

The normalized current responses (*I*/*c*) (response factors) for E1, E2, E3, and EE2 in BR buffer with 33% acetonitrile at pH 9.04 reveal significant differences in their electrochemical oxidizability, reflecting their structural characteristics (Figure 4). E3 exhibits the highest normalized current response, with a distinct peak around 1280 mV. This can be attributed to the presence of one phenolic hydroxyl group and two secondary hydroxyl groups, where the phenolic OH group serves as the primary oxidation site, while the secondary hydroxyl groups may contribute to the overall current to a lesser extent [32].

E2 and E1 show lower responses compared to E3, which aligns with their reduced number of oxidizable sites. For E2, the combination of one phenolic OH group and one secondary OH group provides fewer oxidation sites, leading to a lower current response [32]. In the case of E1, the presence of only one phenolic hydroxyl group makes it the least reactive among the natural estrogens, as it offers only a single oxidation site [32].

In contrast, EE2 demonstrates the lowest normalized current response. This is due to its structural stability, where the tertiary hydroxyl group cannot undergo further oxidation because the carbon atom to which it is attached is fully substituted, making EE2 significantly more resistant to electrochemical oxidation compared to the other estrogens [32].

To enable a comprehensive determination of total estrogenic content, it was necessary to identify a single potential at which the response represented the sum of all four estrogens. By examining the current responses normalized to concentration, it was found that the variability of these normalized signals was minimal at +1220 mV (Figure 4). Thus, the current measured at this potential can be used to represent the total concentration of the estrogen mixture. Standard addition of EE2 was employed to quantify the total estrogen content, as EE2 served as a suitable calibration standard.

Figure 5 demonstrates the voltammetric curves obtained when analyzing mixtures of the four target estrogens, both before and after addition of an EE2 standard.

The results (Table 3) show that the determined total estrogen concentrations deviated by less than 10% from the expected values, which is acceptable given the complexity of the mixture and the simultaneous detection approach.

Repeatability experiments (Table 4) indicated that the method is precise, with differences from the expected values under 4% and a standard deviation of 0.60 µmol·L^−1^.

### 2.5. Application to Real Water Samples

Since the detection limits achieved under the optimized conditions remained higher than those typically reported in natural waters [3], a preconcentration step was integrated. Vacuum evaporation of large sample volumes (0.5–1.0 L) at 55 °C, followed by redissolution in ACN and subsequent addition of BR buffer (pH 9.04), allowed for successful detection of estrogenic compounds.

Model solutions containing EE2 at low concentrations (on the order of 10^−8^ mol·L^−1^) were processed by vacuum evaporation and analyzed using the standard addition method. The results (Table 5) confirmed that this preconcentration step did not affect analyte integrity, as the recovered concentrations matched the spiked values closely.

The procedure was then applied to a real sample of surface water collected from a natural bathing site (Mělice). After filtration and subsequent evaporation, the sample was spiked with 25 µL of EE2 (4.99 × 10^−4^ mol·L^−1^) and 25 µL of E2 (5.76 × 10^−4^ mol·L^−1^), resulting in a total added concentration of 2.68 × 10^−8^ mol·L^−1^. This spiking was performed due to the anticipated low concentration of estrogenic substances in the original sample. The voltammetric oxidation curves (Figure 6) revealed the presence of a complex matrix background, which required mathematical approximation for background subtraction.

Analysis of the spiked sample using the standard addition method for EE2 yielded a measured concentration of 3.16 × 10^−8^ mol·L^−1^, which is higher than the added concentration. The difference between the measured concentration and the spiked concentration corresponds to the natural presence of estrogenic substances in the original water sample. Based on this calculation, the original water sample was determined to contain approximately 4.8 × 10^−9^ mol·L^−1^ of estrogenic substances. This measured concentration falls within the range reported for similar aquatic environments [3], thereby confirming the method’s suitability for real-world environmental monitoring.

## 3. Conclusions

The development and optimization of a DPV-based method for determining estrogenic compounds in water demonstrated that carefully adjusting the supporting electrolyte (BR buffer pH 9.04 with 33% ACN), pH, and organic solvent content leads to reproducible and linear responses for EE2, E1, E2, and E3. By selecting a common potential (+1220 mV) and applying the standard addition method, we achieved reliable estimation of total estrogen content in both model and real environmental samples. Integration of a vacuum evaporation step further expanded the method’s applicability to low-concentration surface water analysis. While HPLC-MS is a powerful technique for the separation and quantification of individual estrogenic compounds, it requires complex and costly sample preparation (e.g., solid-phase extraction or supercritical fluid extraction) and significant operational expenses [33,34,35]. In contrast, the presented voltammetric method focuses on the cumulative determination of estrogens, offering a simple, cost-effective, and rapid approach that minimizes sample preparation and is well suited for routine applications. Given the different objectives of these methods, direct comparison may not be entirely meaningful, as each addresses unique analytical needs. Future efforts may focus on lowering detection limits through electrode surface modifications or online preconcentration, thereby broadening the method’s scope for routine environmental monitoring.

## 4. Materials and Methods

### 4.1. Reagents and Chemicals

Analytical standards of estrone (E1), 17β-estradiol (E2), estriol (E3), and 17α-ethinylestradiol (EE2), each with a purity of ≥98%, were obtained from Merck KGaA (Darmstadt, Germany). Organic solvents, such as ethanol and acetonitrile (both ≥ 99.8%), were also purchased from Merck. Britton–Robinson (BR) buffer solutions with pH values ranging from 2.1 to 11.19 were prepared and used as the supporting electrolyte. The pH was adjusted by mixing acidic (H_3_PO_4_, 85%, Lachema, p.a.; CH_3_COOH, 99.8%, Penta, p.a.; H_3_BO_3_, Penta, p.a.) and alkaline (0.2 mol L^−1^ NaOH) components. Milli-Q^®^ purified water with an electrical resistivity lower than 18.3 MΩ·cm was used throughout the experiments.

### 4.2. Instrumentation

Voltammetric measurements were carried out using an EP 100VA electrochemical analyzer (HSC servis, Bratislava, Slovakia). The electrochemical cell consisted of a three-electrode system, including a boron-doped diamond electrode (BDDE) as the working electrode (BioLogic, Seyssinet-Pariset, France), with an active surface area of 7.07 mm^2^, an inner diameter of 3.0 mm, and a B/C ratio of 1000 ppm. The reference electrode was a saturated Ag|AgCl|KCl electrode, and the auxiliary electrode was a platinum wire (Monokrystaly s.r.o., Turnov, Czech Republic). The pH of the buffer solutions was measured using an ORION STAR A221 pH meter (Thermo Fisher Scientific, Waltham, MA, USA).

### 4.3. Voltammetric Measurements

Differential pulse voltammetry (DPV) was employed for the electrochemical detection of the four estrogens. Measurements were performed in a solution consisting of 10 mL of BR buffer and 5 mL of acetonitrile, adjusted to pH 9.00. The optimized parameters for DPV were as follows: potential range from +0.7 V to +2.0 V, scan rate of 25 mV s^−1^, pulse amplitude of 30 mV, and pulse time of 60 ms. Reproducibility of the response on the BDDE was ensured through electrochemical treatment, where the electrode was polarized for 10 s at potentials of +2000 mV, −200 mV, and +2000 mV consecutively. Each measurement was repeated a minimum of five times to ensure reproducibility.

### 4.4. pH Study and Supporting Electrolyte Composition

The influence of pH on the electrochemical behavior of 17α-ethinylestradiol (EE2), estrone (E1), 17β-estradiol (E2), and estriol (E3) was investigated using Britton–Robinson buffer over a pH range from 2.04 to 10.00. The effect of acetonitrile concentration in the supporting electrolyte was also studied, with acetonitrile content varying from 0% to 66%. At lower pH values, the oxidation peaks occurred at more positive potentials, and the current response exhibited significant nonlinearity. As the acetonitrile concentration increased, the oxidation peaks shifted to more positive potentials, with better linearity observed between current response and concentration.

### 4.5. Calibration and Quantification

For quantification of the estrogen mixtures, a standard addition method was employed using 17α-ethinylestradiol (EE2) as the reference compound. Calibration curves were constructed using Origin 9.0 software, which applies the method of least squares to determine the relationship between the current response and concentration. The linear range for the quantification of estrogens was found to be between 15.35 and 134.55 µmol L^−1^.

## Figures and Tables

**Figure 1 molecules-30-00751-f001:**
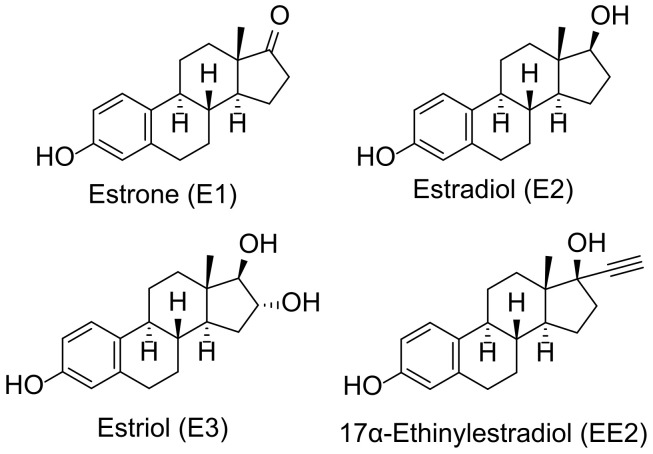
Structures of the most common estrogenic compounds.

**Figure 2 molecules-30-00751-f002:**
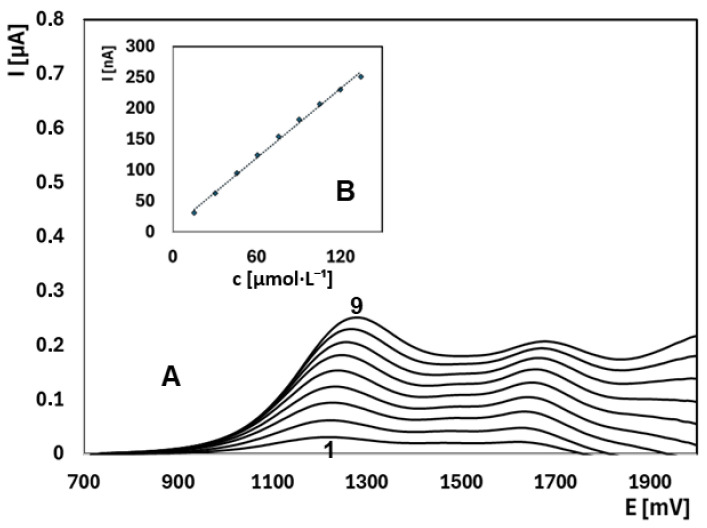
Anodic oxidation curves of EE2 in BR buffer with 33% acetonitrile (pH 9.04) after background subtraction. A: Corrected curves; B: Current vs. concentration (15.35–134.55 µmol·L^−1^). Curves: 1—15.35 µmol·L^−1^, 9—134.55 µmol·L^−1^. Parameters: initial potential +700 mV, final +2000 mV, scan rate 25 mV·s^−1^, pulse amplitude 30 mV, pulse duration 60 ms.

**Figure 3 molecules-30-00751-f003:**
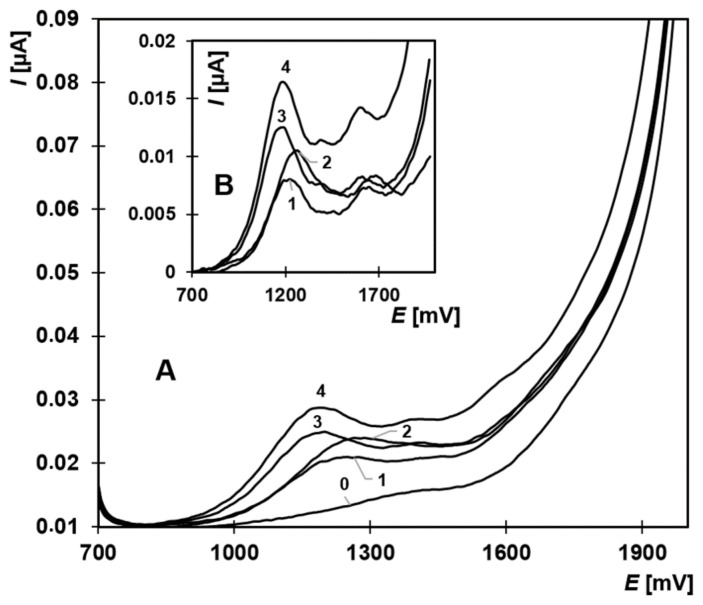
Anodic oxidation curves of E1, E2, E3, and EE2 in BR buffer with 33% acetonitrile (pH 9.04). A: Original measurements; B: Background-corrected oxidation curves. Curves: 0—supporting electrolyte, 1—EE2 (6.66 µmol·L^−1^), 2—E3 (2.13 µmol·L^−1^), 3—E1 (5.34 µmol·L^−1^), 4—E2 (7.68 µmol·L^−1^). Parameters: initial potential +700 mV, final +2000 mV, scan rate 25 mV·s^−1^, pulse amplitude 30 mV, pulse duration 60 ms.

**Figure 4 molecules-30-00751-f004:**
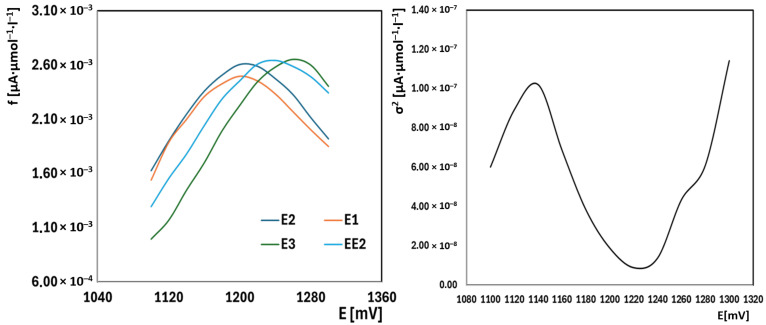
Left: Response factors (f) of compounds E1, E2, E3, and EE2; Right: Analysis of the variance in response factors (f) for compounds E1, E2, E3, and EE2.

**Figure 5 molecules-30-00751-f005:**
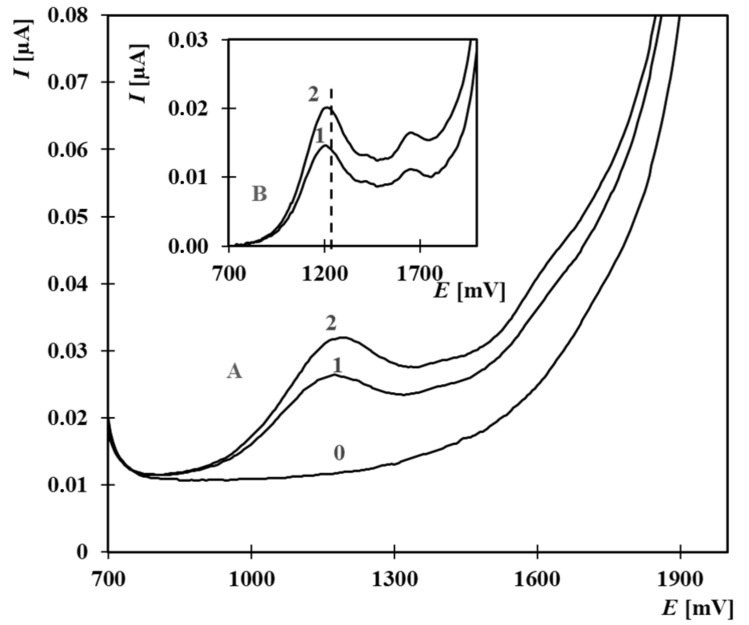
Anodic oxidation curves for estrogens (1.34 µmol·L^−1^ E1, 3.84 µmol·L^–1^ E2, 0.53 µmol·L^−1^ E3, 3.33 µmol·L^−1^ EE2) using standard addition of EE2 (3.33 µmol·L^−1^). A—Original curves, B—Background-subtracted. Curves: 0—electrolyte, 1—estrogen mixture, 2—EE2 addition. Electrolyte: BR buffer (pH 9.04) with 33% acetonitrile. Parameters: +700 to +2000 mV, 25 mV·s^−1^, 30 mV amplitude, 60 ms pulse.

**Figure 6 molecules-30-00751-f006:**
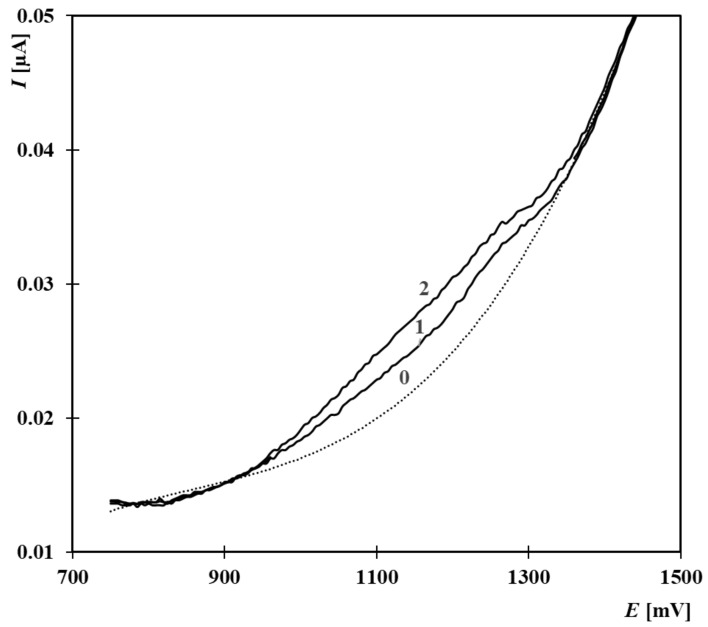
Anodic oxidation curves obtained from the analysis of a real surface water sample from the natural swimming area in Mělice. Individual curves: 0—approximated background curve, 1—sample curve with a spiked concentration of 2.68 × 10^−8^ mol·L^−1^ E2 and EE2, 2—standard addition of EE2 (1.66 µmol·L^−1^). Supporting electrolyte: BR buffer with pH 9.04 and 33% acetonitrile. Measurement parameters: initial potential +700 mV, final potential +2000 mV, polarization rate 25 mV·s^−1^, pulse amplitude 30 mV, and pulse duration 60 ms.

**Table 1 molecules-30-00751-t001:** Results of EE2 determination in three basic electrolytes at pH 9.04.

Composition of the Supporting Electrolyte	Concentration Range [µmol·L^−1^]	Calibration Equation	R^2^
15 mL BR	15.35–105.33	*I* = 1.88*c* − 6.76	0.998
10 mL BR + 5 mL ACN	15.35–134.55	*I* = 1.90*c* + 6.38	0.999
5 mL BR + 10 mL ACN	15.35–134.55	*I* = 1.75*c* + 11.11	0.998

**Table 2 molecules-30-00751-t002:** Current–concentration relationships for 17α-ethinylestradiol oxidation (15.35–134 µmol/L) at various pH values in BR buffer with 33% acetonitrile.

pH	Calibration Equation	R^2^
2.04	*I* = −0.003*c^2^ +* 1.07*c* − 1.73	0.9995
4.03	*I* = −0.0024*c^2^* + 3.77*c* + 2.47	0.9980
6.06	*I* = −0.008*c^2^ +* 2.77*c* − 10.50	0.9911
9.04	*I* = 1.86*c* + 8.39	0.9984
10.00	*I* = 1.78*c* + 8.87	0.9956

**Table 3 molecules-30-00751-t003:** Comparison of analyzed and found estrogen concentrations.

Analyzed [µmol·L^−1^]					Found [µmol·L^−1^]	Δ [%]
E1	E2	E3	EE2	Σ Estrogens	Σ Estrogens	
1.34	1.92	0.53	1.66	5.45	5.95	+9.17
1.34	3.84	0.53	3.33	9.03	8.47	−6.28
1.34	1.92	0.53	3.33	7.06	7.12	+0.85
1.87	2.69	0.74	3.33	8.63	8.73	+1.37
2.68	3.84	1.06	3.33	10.90	11.14	+2.11

**Table 4 molecules-30-00751-t004:** Repeatability of total estrogen determination.

No.	Found Σ Estrogens [µmol·L^−1^]	Δ [µmol·L^−1^]	Δ [%]
1	11.24	+0.33	+3.02
2	11.07	+0.16	+1.47
3	10.98	+0.07	+0.64
4	10.95	+0.04	+0.37
5	10.93	+0.02	+0.18
Mean: 11.03 µmol·L^−1^ SD: 0.60 µmol·L^−1^ CI: 11.03 ± 0.74 µmol·L^−1^			

**Table 5 molecules-30-00751-t005:** Results of analysis of model samples containing low concentrations of EE2 using the selected concentration procedure.

No.	Sample	Analyzed [mol·L^−1^]	Found [mol·L^−1^]	Δ [mol·L^−1^]	Δ [%]
1	Model	9.99 × 10^–8^	9.99 × 10^–8^	0	0
2	Model	2.47 × 10^–8^	2.54 × 10^–8^	+0.07	+2.83

## Data Availability

Data are contained in the article.

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
