# Peer review of "Voltammetric Determination of the Total Content of the Most Commonly Occurring Estrogens in Water Media"

_molecules, 2025, doi:10.3390/molecules30030751_

Round 1

Reviewer 1 Report

Comments and Suggestions for Authors

In this paper, the authors used a cost-effective and straightforward method for  simultaneous determination of four estrogens from water. The article is especially addressed to analytical chemists and environmental experts.

The abstract is short but informative, presenting all necessary details in order to understand the purpose of this paper.

Introduction presents the state of the art in this area, presenting the potential and impact of estrogens in the environment and human health. It is explained why they have chosen a new voltammetric method for estrogen determination in water.

The novelty of this paper consists in the simultaneous determination of four estrogens.

Differential pulse voltammetry was employed for the electrochemical detection. In experimental part are given all details in order to understand how this method was employed.

Result and discussion present and explain in detail the results obtained.

So authors showed that this method for determining estrogenic compounds in water demonstrated that carefully adjusting the supporting electrolyte (BR buffer pH 9.04 with 33% ACN), pH, and organic solvent content leads to reproducible and linear responses for those estrogens.

Even this paper is short, it is well-documented and written.

Author Response

Dear Reviewer,

Thank you for your time and effort in reviewing our manuscript. We greatly appreciate your evaluation and are glad that you found our work satisfactory.

Reviewer 2 Report

Comments and Suggestions for Authors

The article entitled  "Voltammetric Determination of the Total Content of the Most Commonly Occurring Estrogens in Water" is focused on the voltammetric determination of four prevalent estrogens estrone, 17β-estradiol, estriol and ethinylestradiol—and presents optimized conditions for their simultaneous quantification.  The novelty of this work is development and optimization of a DPV-based method for determining estrogenic compounds. Also, the main parameters affecting the response of electrode to estrogens have been examined. The authors achieved reliable estimation of total estrogen content in model sample and determined the estrogen content in water sample after pre-treatment. From practical aspect, the discussed voltammetric approach simplifies sample preparation, reduces costs and it seems to be a good alternative to conventional HPLC-MS techniques.  Overall, the work is good written, but some points should be clarified.

1.      The working electrode, its manufacturer, the reasons for its selection, the range of working potentials should be described in the work. Please add this information to the text of the article.

2.      The title of the article suggests the determination of estrogens in water, but it seems that this cannot be done directly without modification. In this case, the question arises about the correctness of the title of the work. It would be better to use « Voltammetric Determination of the Total Content of the Most Commonly Occurring Estrogens in Water media».

3.      The CV curves of the electrode are presented only in the anodic region, and what is observed in the cathodic region.

4.      Did the stability of the response from the number of CV cycles was studied? Why did only the signal from the first peak processed for quantitative assessment of estrogen content?

5.      There is only one report [Chemosphere 2021, 267, 128888, doi:10.1016/j.chemosphere.2020.128888] presented in the Result and Discussion section. Please compare your results with data from other methods (HPLC-MS) and expand this section.

Author Response

Dear Reviewer,

Thank you for your valuable and constructive feedback. Below are our responses to your specific comments:

  1. “The working electrode, its manufacturer, the reasons for its selection, the range of working potentials should be described in the work. Please add this information to the text of the article.”
    We have added a justification for the use of BDDE in the introduction and provided detailed information in the experimental section.
  2. “The title of the article suggests the determination of estrogens in water, but it seems that this cannot be done directly without modification. In this case, the question arises about the correctness of the title of the work. It would be better to use « Voltammetric Determination of the Total Content of the Most Commonly Occurring Estrogens in Water media».”
    We agree and have updated the title accordingly.
  3. “The CV curves of the electrode are presented only in the anodic region, and what is observed in the cathodic region.”
    We did not directly measure CV curves in the cathodic region, but it is well-documented in the literature (10.1016/j.talanta.2020.121804) that no peaks or processes occur there.
  4. “Did the stability of the response from the number of CV cycles was studied? Why did only the signal from the first peak processed for quantitative assessment of estrogen content?”
    CV stability was not specifically studied; however, electrochemical treatment was performed before each use, as described in the experimental section. The first peak is less prone to error and thus more reliable for quantitative evaluation.
  5. “There is only one report [Chemosphere 2021, 267, 128888, doi:10.1016/j.chemosphere.2020.128888] presented in the Result and Discussion section. Please compare your results with data from other methods (HPLC-MS) and expand this section.”
    We agree and added a citation to a recent studies. However, we believe that HPLC is a significantly more expensive and complex technique with greater sample preparation requirements, making a direct comparison less relevant.

We hope these revisions address your concerns and improve the manuscript. Thank you for your valuable suggestions.

Reviewer 3 Report

Comments and Suggestions for Authors

1.      Table 2 needs to provide the relevant linear coefficients under different pH conditions.

Author Response

Dear Reviewer,

Thank you for your helpful suggestions. Please see our response below:

  1. “Table 2 needs to provide the relevant linear coefficients under different pH conditions.”
    We agree and have included the requested data.

We appreciate your feedback and hope these changes enhance the manuscript.